# Responses of Nitrous Oxide Emissions and Bacterial Communities to Experimental Freeze–Thaw Cycles in Contrasting Soil Types

**DOI:** 10.3390/microorganisms11030593

**Published:** 2023-02-26

**Authors:** Wenyan Li, Peter Semba Mosongo, Wenxu Dong, Arbindra Timilsina, Ruibo Sun, Fenghua Wang, Anna Walkiewicz, Binbin Liu, Chunsheng Hu

**Affiliations:** 1Key Laboratory of Agricultural Water Resources, Hebei Key Laboratory of Soil Ecology, Center for Agricultural Resources Research, Institute of Genetics and Developmental Biology, Chinese Academy of Sciences, 286 Huaizhong Road, Shijiazhuang 050021, China; 2University of Chinese Academy of Sciences, No. 19 (A) Yuquan Road, Shijingshan District, Beijing 100049, China; 3Xiong’an Institute of Innovation, Chinese Academy of Sciences, Xiong’an New Area 071700, China; 4Anhui Province Key Laboratory of Farmland Ecological Conservation and Pollution Prevention, Key Laboratory of JiangHuai Arable Land Resources Protection and Eco-Restoration, College of Resources and Environment, Anhui Agricultural University, No. 130, West Changjiang Road, Hefei 230036, China; 5Hebei Key Laboratory of Environmental Change and Ecological Construction, Hebei Experimental Teaching Demonstrating Center of Geographical Science, School of Geographical Sciences, Hebei Normal University, Shijiazhuang 050024, China; 6Institute of Agrophysics, Polish Academy of Sciences, Wiadczalna 4, 20-290 Lublin, Poland

**Keywords:** freeze–thaw cycles, denitrification potential, N_2_O, upland soil, bacterial community structure

## Abstract

Nitrous oxide (N_2_O) pulse emissions are detected in soils subjected to freeze–thaw cycles in both laboratory and field experiments. However, the mechanisms underlying this phenomenon are poorly understood. In this study, a laboratory incubation experiment that included freeze–thaw cycles (FTC), freezing (F) and control (CK) treatments was performed on three typical Chinese upland soils, namely, fluvo-aquic soil (FS), black soil (BS) and loess soil (LS). A higher similarity in soil properties and bacterial community structure was discovered between FS and LS than between FS and BS or LS and BS, and the bacterial diversity of FS and LS was higher than that of BS. FTC significantly increased the denitrification potential and the proportion of N_2_O in the denitrification gas products in FS and LS but decreased the denitrification potential in BS. Accordingly, with the increasing number of freeze–thaw cycles, the bacterial community composition in the FTC treatments in FS and LS diverged from that in CK but changed little in BS. Taxa that responded to FTC or correlated with denitrification potential were identified. Taken together, our results demonstrated that the effects of FTC on N_2_O emissions are soil-type-dependent and that the shift in the microbial community structure may contribute to the elevated N_2_O emissions.

## 1. Introduction

Soil freeze–thaw cycles are a common phenomenon in high-altitude regions, and the intensity and frequency of freeze–thaw cycles are primarily dependent on the regional climate [1]. Approximately 55% of the total land area of the Northern Hemisphere experiences seasonal freezing [2]. Prior studies have demonstrated that freeze–thaw cycles can strongly affect soil physiochemical properties, microbial community composition and activity [3,4,5]. Freeze–thaw cycles increase nutrient availability for soil microorganisms and alter soil biochemical processes [6], presumably due to the death of microbes and the disruption of soil aggregates [7].

Approximately 40–70% of annual N_2_O emissions from agricultural soils in temperate regions were reported in the thawing period of the beginning of spring [8]. Current knowledge suggests several possible mechanisms for N_2_O emissions during the thawing process. One is the physical release of N_2_O that has been trapped in the frozen layers [9], and this notion was supported by several studies in which soil N_2_O accumulation in winter was orders of magnitude higher than ambient concentrations [10]. N_2_O could also be produced by de novo processes at the beginning of thawing, which might be due to increased biological activity [11]. Evidence supporting this observation included the increase in the diversity of nitrifying and denitrifying bacteria during the N_2_O emissions thaw event [12]. The increase in soil carbon and nitrogen substrates during freeze–thaw cycles may also contribute to N_2_O emissions, and the priming effect could play a role during these processes [13,14]. A recent study with lysimeters suggested that the effects of freeze–thaw events on the nitrogen loss routes and N_2_O fluxes were dependent on soil texture [15].

N_2_O emissions lead to soil nitrogen loss and intensify greenhouse effects [16]. The atmospheric N_2_O mixing ratio has increased by more than 20% since the industrial revolution, and the N_2_O emissions from agricultural activities have increased sharply in the past few decades [17,18]. Nitrification and denitrification are considered two key biochemical processes that contribute to N_2_O emissions in terrestrial ecosystems [19], while soil denitrification has been reported as the dominant process responsible for N_2_O emissions in farmland soil [20,21]. Most studies of freeze–thaw N_2_O emissions have concluded that denitrification is the dominant process responsible for large N_2_O fluxes [22,23]. N_2_O emissions can be altered by soil management practices such as fertilization, and the effects were reported to be strongly dependent on soil type [24,25]. A large variation was also discovered in the absolute amount of N_2_O emitted from different arable soils during freeze–thaw events [9], suggesting that the effects of freeze–thaw cycles on N_2_O emissions are strongly associated with the soil properties.

The soil bacterial community is the dominant driver of the denitrification process. Soil N_2_O and CO_2_ emissions have been linked with cultivable bacterial populations [26,27]. A number of studies have been conducted to investigate the microbial responses to freeze–thaw events, but the results were not consistent. Several early studies using temperature/denaturing gradient gel electrophoresis (TGGE/DGGE) did not observe clear changes in the microbial community structure during freeze–thaw events [28,29,30]. Recent studies using high-throughput sequencing technology revealed changes in the microbial community composition due to freeze–thaw cycles [31].

In this study, we used three soils under conventional fertilization: fluvo-aquic soil, loess soil and black soil from Hebei Province, Shaanxi Province and Jilin Province of China, respectively. We analyzed the effects of freeze–thaw cycles on the soil denitrification potential and N_2_O emissions, and tried to identify the factors contributing to these effects from the perspective of soil bacterial community structure.

## 2. Materials and Methods

### 2.1. Soils

The three types of soil used in this study were fluvo-aquic soil (FS), black soil (BS) and loess soil (LS). FS and LS are classified as Cambic Arenosols, and BS is classified as a Haplic Chernozems according to the Food and Agriculture Organization of the United Nations (FAO) [32]. FS samples were collected at the Luancheng Agroecosystem Experiment Station of the Chinese Academy of Sciences in Shijiazhuang, Hebei Province, China (37°53′N, 114°41′E, elevation 50 m). BS samples were collected at Gongzhuling, Jilin Province, China (43°51′N, 124°82′E, elevation 300 m). LS samples were collected at Changwu Agro-Ecological Experimental Station in Shaanxi Province, China (35°28′N, 107°88′E, elevation 1200 m). The 0–10 cm topsoil samples were collected from the same plot with three replicates in July 2020, and the three sampling sites were conventionally fertilized cropland by local farmers. Soil samples were sieved through a 2 mm mesh to remove plant residue and other impurities and then homogenized thoroughly in the laboratory. Each soil sample was divided into three parts: one part was stored at −80 °C for total DNA extraction, the second part was stored at 4 °C for the culture experiment and the determination of soil properties, including soil ammonium nitrogen (NH_4_^+^-N), nitrate nitrogen (NO_3_^−^-N), organic carbon (SOC), dissolved organic carbon (DOC), dissolved organic nitrogen (DON) concentrations and soil pH, and the third part was air-dried at room temperature to determine soil total nitrogen (TN) and total carbon (TC) concentrations.

The soil moisture content was determined using the gravimetric method. Soil pH was measured using deionized carbon dioxide-free water at a ratio of 1:5 (g: mL) with a pH meter (PHS-3C; Shanghai INESA, Shanghai, China). Soil NH_4_^+^-N and NO_3_^−^-N were extracted using 50 mL of 1 mol L^−1^ KCl solution from 10 g of fresh soil and were measured using a Smartchem 140 automatic analyzer (AMS/Westco, Rome, Italy) and dual-wavelength ultraviolet spectrophotometer (UV-2450; SHIMADZU, Suzhou, China), respectively. SOC was determined using the potassium dichromate oxidation method. Soil TC and TN were measured using a CHNOS elemental analyzer (Vario MAX; Elementar, Hanau, Germany) after the soil was air-dried and ground. Ten grams of fresh soil was mixed with 50 mL of 0.5 mol L^−1^ K_2_SO_4_ and shaken for 2 h at 25 °C, and the filtrate was used to determine soil DOC using a Liqui TOCII analyzer (Elementar, Hanau, Germany). DON was determined using an ultraviolet spectrophotometer (UV-2450; SHIMADZU, Suzhou, China) after the filtrate was sterilized at 105 °C for half an hour under alkaline conditions and then treated with HCl.

### 2.2. Incubation Experiment and Gas Measurements

Three treatments were carried out in this study, and samples in each treatment were incubated for 10 days: (1) for the freeze–thaw cycles (FTC), the soils underwent FTC, and each freeze–thaw cycle included an incubation at −10 °C for 12 h followed by 10 °C for 12 h; (2) for the freezing treatment (F), the soils were frozen at −10 °C and (3) for the control treatment (CK), the soils were incubated at 10 °C. Each 120 mL serum flask was filled with 20 g of sieved soil, and the soil moisture content was adjusted to 60% of the field capacity by adding distilled water. The flasks were then sealed with butyl rubber and aluminum caps. The serum flasks were evacuated to less than 0.1 kPa and flushed with pure helium (99.999%) five times to make the headspace environment free of oxygen (O_2_ concentration < 450 ppm) and dinitrogen (N_2_). In each treatment, 5 replicate serum flasks were used for measuring the concentration of accumulated gases N_2_, N_2_O and CO_2_, and 15 serum flasks were opened at five incubation times (2nd, 4th, 6th, 8th and 10th day) for extracting soil DNA and for determining the soil NH_4_^+^-N and NO_3_^−^-N concentrations. The cumulative concentrations of CO_2_, N_2_O and N_2_ in the flasks were measured every 24 h using a robotized sampling and analysis system, which consisted of an autosampler (CTC PAL, Zwingen, Switzerland), a peristaltic pump (Gilson minipuls 3; LeBel, France) and a gas chromatograph (Agilent 7890B; Santa Clara, CA, USA). The N_2_O concentration was determined using a thermal conductivity detector (TCD) (N_2_O concentration ≥ 5 ppm) and an electron capture detector (ECD) (N_2_O concentration < 5 ppm), and the N_2_, CO_2_ and O_2_ concentrations were determined using TCD. The details of this system and the calculation methods of the gas concentrations were described by Molstad et al. [33]. The soil CO_2_-C (mg C kg^−1^ dry soil d^−1^), N_2_O-N (mg N kg^−1^ dry soil d^−1^) and N_2_-N (mg N kg^−1^ dry soil d^−1^) production rates were the rates of anaerobic culture for 10 days under the FTC, F and CK treatments. In this study, the soil denitrification potential was defined as denitrification activity under anaerobic conditions and was calculated as the sum of the N_2_-N and N_2_O-N production rates [34].

### 2.3. Quantification of N-Cycling Genes

Soil total DNA was extracted from 0.5 g of soil using the FastDNA Spin Kit for Soil (MP Biomedicals, Santa Ana, CA, USA) following the manufacturer’s instructions. The concentration and purity of the DNA were assessed using a NanoDrop One spectrophotometer (Thermo Fisher Scientific, Madison, WI, USA). The abundances of the archaeal ammonium monooxygenase gene (AOA *amoA* gene), bacterial ammonium monooxygenase gene (AOB *amoA* gene), the nitrite reductase gene (*nirK* and *nirS* gene) and the N_2_O reductase gene (*nosZ* gene) were quantified using a qPCR machine (Bio-Rad, CFX, Hercules, CA, USA), and the primers used in this study were Arch-amoAF/Arch-amoAR, amoA-1F/amoA-2R, nirK-F1aCu/nirK-R3Cu, nirS-R3cd/nirS-cd3aF and nosZ-F/nosZ-1622R, respectively [35,36,37]. The amplification system of the target genes was a 20 µL mixture containing 10 µL 2 × TB Green^TM^ Premix EX Taq^TM^ (Tli RNase H Plus) (Takara Biotechnology, Dalian, China), 0.5 µL (10 µM) of each primer, 1 µL DNA template (diluted to 20 to 30 ng µL^−1^) and 8 µL sterile double-distilled water (ddH_2_O). Standard curves were constructed using 10-fold dilutions of plasmids carrying the respective target gene fragments. Details about the primers and amplification procedures were described in previous studies [38].

### 2.4. Amplicon Sequencing and Bioinformatics Analysis

The V4 regions of the bacterial 16S rRNA gene were amplified using the primer sets 515F (5′-GTGYCAGCMGCCGCGGTAA-3′) and 806R (5′-GGACTANVGGGTWTCT-AAT-3′) to investigate the soil bacterial community structure and diversity using high-throughput amplicon sequencing technology on an Illumina NovaSeq platform (Illumina, San Diego, CA, USA) [39]. Bioinformatics analysis was performed mainly using QIIME2 (version 2020.11, https://qiime2.org/, accessed on 6 February 2023) [40]. First, the amplicon sequences were imported into the QIIME2 environment, and primer removal, quality control and homologous sequence clustering were performed using the DADA2 pipeline [41]. The taxonomic classification of the represented feature sequences was performed through alignment with the SILVA 138 reference database [42,43]. Sequences that were not identified as bacteria were deleted. For normalization, sequence counts of all samples were rarefied to the number of the sample with the lowest total read count. The raw sequencing data were deposited in the National Center for Biotechnology Information (NCBI) under BioProject ID PRJNA792883.

### 2.5. Statistical Analysis

Analysis of variance (ANOVA) was conducted to determine if there were statistically significant differences in the soil denitrification potential, gene abundances and relative abundance of bacterial genera between the FTC, F and CK treatments using the R package “agricolae” [44]. Pearson correlation analysis was performed for gene abundances, bacterial taxa and gas production rates using the R package “psych” [45]. A Venn diagram was constructed in R using the package “VennDiagram” [46].

## 3. Results

### 3.1. Soil Properties

The investigated chemical properties of the three soils were quite different from each other (Table 1). BS is an acidic soil, while FS and LS are alkaline soils. The TC, TN, SOC, DON and NO_3_^−^-N were highest in FS. The DOC contents of BS and FS were significantly higher than that of LS. The NH_4_^+^-N concentration in BS was significantly higher than that in FS and LS. The NO_3_^−^-N contents of FS, BS and LS were 47.84, 19.56 and 11.40 mg kg^−1^ dry soil, respectively. Inorganic nitrogen mainly existed in the form of nitrate (NO_3_^−^-N) in the three soils. The C/N ratio (TC/TN) of LS was the highest, followed by FS and BS. The abundances (copy numbers g^−1^ dry soil) of the nitrification and denitrification functional genes in FS were significantly higher than those in LS, except that the *nosZ* gene was not significant; the BS were lowest (the CK samples of day 0 in Figure 1).

In total, 9579 amplicon sequence variants (ASVs) were detected in the three original soils. There were 3990 ASVs in FS, 2588 ASVs in BS and 4459 ASVs in LS. FS and LS had 1278 shared ASVs, FS and BS had 119 shared ASVs, and BS and LS had 148 shared ASVs (Figure 2a). The Pearson correlation based on ASV abundances among the three original soils showed that the similarity in the bacterial community structure was higher between FS and LS than between FS and BS or LS and BS (Figure 2b). The bacterial community was constituted by 14 dominant phyla with a relative abundance greater than 1%, and the four phyla with the highest relative abundance were Actinobactera (FS 23.62%, BS 44.85%, LS 22.18%), Proteobacteria (FS 21.61%, BS 25.46%, LS 17.80%), Acidobacteria (FS 15.86%, BS 9.14%, LS 20.49%) and Chloroflexi (FS 10.48%, BS 7.24%, LS 12.42%) (Appendix A). The top 20 bacterial genera with the highest relative abundance in Appendix A show the differences in bacterial community composition among the three original soils.

### 3.2. Effects of Freeze–Thaw Cycles on Soil Denitrification

In the CK treatment, the soils were anaerobically incubated at 10 °C for 10 days. The soil denitrification potential and the N_2_O and N_2_ production rates of FS were higher than those of BS and LS. BS had the highest N_2_O/(N_2_O + N_2_) ratio, followed by FS and LS (Figure 3a). The soil denitrification potential and the N_2_ and N_2_O production rates were positively correlated with the initial soil nitrification and denitrification functional gene abundance (Appendix A). In the CK treatment, the soil NH_4_^+^-N contents fluctuated a little, while the soil NO_3_^−^-N contents decreased during the incubation (Figure 4). The *nirK*, *nirS* and *nosZ* gene abundances showed an increasing trend, and the AOA *amoA* and AOB *amoA* gene abundances varied little during incubation in the CK treatment in the three types of soils (Figure 1).

Compared with the CK treatment, the FTC and F treatments altered the soil NH_4_^+^-N and NO_3_^−^-N contents and gas production rates of all three soils (Figure 3a and Figure 4). In FS and LS, the FTC treatment significantly increased the soil denitrification potential, CO_2_ and N_2_O production rates and the N_2_O/(N_2_O + N_2_) ratio. The F treatment increased the N_2_O production rate and N_2_O/(N_2_O + N_2_) ratio in FS but not in LS. In BS, the FTC and F treatments significantly decreased the soil denitrification potential and the CO_2_, N_2_O and N_2_ production rates. Significant positive linear correlations between the CO_2_ production rate and soil denitrification potential among the different treatments in the three soils were observed (R_FS_ = 0.885, P_FS_ < 0.001; R_BS_ = 0.984, P_BS_ < 0.001; R_LS_ = 0.629, P_LS_ < 0.001) (Figure 3b). The N_2_O/(N_2_O + N_2_) ratio was positively correlated with the CO_2_ production rate among the different treatments in FS (R = 0.812, *p* < 0.001) and LS (R = 0.503, *p* < 0.001), while the correlation was negative (R = −0.364, *p* < 0.001) in BS (Figure 3c). In FS and LS, compared with the CK treatment, the soil NH_4_^+^-N content in the FTC treatment significantly increased, while the NH_4_^+^-N content in the F treatment only changed a little. In BS, the FTC treatment increased the soil NH_4_^+^-N content, while F decreased the soil NH_4_^+^-N content compared with CK. Under anaerobic culture conditions, the NO_3_^−^-N contents of the three types of soils decreased dramatically in all treatments. The soil NO_3_^−^-N contents of FS and BS in the FTC treatment, and the NO_3_^−^-N content of BS in the F treatment significantly increased compared with those in the CK treatment (Figure 4).

### 3.3. Effects of Freeze–Thaw Cycles on the Bacterial Community Structure and Diversity

The response of dominant bacterial genera to the FTC or F treatment was illustrated by calculating the log response ratio (LRR) in the three soil types (Figure 5). Different response patterns of the same taxa were discovered among the soils. The relative abundance of *Blastococcus* increased under the FTC treatment in FS and LS but decreased in BS. The Pearson correlation between the soil denitrification potential and the relative abundance of bacterial genera was calculated, and genera that were positively correlated with the denitrification potential at the five time points were identified. *Gaiella* and Un.Solirubrobacterales were positively correlated with the denitrification potential in FS and LS (Figure 5), the two soils in which the denitrification potential and N_2_O production rates were markedly elevated under the FTC treatment (Figure 3a). The PCoA based on the Bray–Curtis distance revealed that FTC and F strongly influenced the bacterial community composition of FS and LS, and the bacterial community diverged with the incubation time, while the effect on BS was small compared with the CK treatment (Figure 6).

## 4. Discussion

The three soils showed great differences in the soil properties and bacterial community structure (Table 1, Figure 2). As all three soil types are upland soils, NO_3_^−^-N is the dominant form of soil inorganic nitrogen. In the CK treatment where FTC were not applied, the NH_4_^+^-N content and nitrification functional gene abundances of the three soils fluctuated little, while the NO_3_^−^-N content decreased considerably and the denitrification functional gene abundances demonstrated an increasing trend with incubation time (Figure 1 and Figure 4). These results likely suggest that denitrification with NO_3_^−^-N as the substrate was a main nitrogen-cycling microbial process after the soils were subjected to anaerobic conditions in this study. In addition, in the FTC treatment, although the nitrogen loss in FS through denitrification increased (Figure 3a), higher concentrations of inorganic nitrogen (NH_4_^+^-N and NO_3_^−^-N) were still observed compared with the CK treatment (Figure 4), suggesting that there were other sources of inorganic nitrogen. One possible source could be nitrogen mineralization, as suggested in previous studies where increased N_2_O emissions and increased soil inorganic nitrogen concentrations were observed together with the promoted soil nitrogen mineralization [6,47].

The CO_2_ production rate reflects microbial respiration and activities. Heterotrophic denitrification with carbon as an electron donor could lead to the production of denitrification products (N_2_ and N_2_O) and CO_2_ [48]. In our study, a positive relationship was observed between the soil total denitrification rate and the anaerobic CO_2_ production rate (Figure 3b), suggesting that heterotrophic denitrification was the dominant process of N_2_O and N_2_ formation, and a similar positive relationship was observed in a previous study [49]. The N_2_O/(N_2_O + N_2_) ratio is used to gain insight into the amount of complete and incomplete denitrification [50]. Positive relationships between CO_2_ emissions and the N_2_O/(N_2_O + N_2_) ratio were observed in FS and LS, while a negative relationship was discovered for BS, suggesting that the underlying mechanisms are soil-type-dependent (Figure 3c).

Functional gene abundances have been widely linked to soil properties and microbial process rates [51]. In the current study, the differences in the abundance of functional genes between the FTC, F and CK treatments were much smaller than the differences between soil types, suggesting that the influence of soil type on nitrogen cycling microbial processes is more critical than that of the FTC treatment (Figure 1). The gene abundances generally varied little under the FTC and F treatments compared with the CK treatment, and this result does not explain the dramatic differences in denitrification potential between the different treatments (Figure 3a). One possible reason could be that FTC and F altered the composition of the functionally active microorganisms, while the total abundance of the functional community remained unchanged. This notion was supported by the LRR analysis, where dramatic changes in the abundance of bacterial genera were discovered between treatments (Figure 5). A change in the composition of the functional nitrogen-cycling microbial taxa while the abundance of the corresponding functional genes was unchanged was also discovered in a previous study where metagenomic analysis was performed to inspect the responses of nitrogen-cycling microorganisms to long-term fertilization [52].

The changes in microbial community composition in response to FTC and F do not rule out the possibility that these treatments also altered microbial functions by regulating the expression of functional genes. Previous studies in Haplic Cambisol grassland soil suggest that freeze–thaw cycles have little impact on nitrifier and denitrifier abundance, but the effects are mainly exerted at the gene expression level [53]. During the freeze–thaw cycles, the soil underwent drastic and complex physical and chemical changes, which could seriously affect the activities of microorganisms and enzymes [54,55]. Although the soil moisture content was adjusted to 60% of the field water capacity in this experiment, the water availability and gas diffusivity may be discrepant due to the differences in natural soil physical properties, which may cause the differences in denitrification potential changes of the three types of soils under FTC. Future studies are needed to illustrate the mechanisms underlying the differences in denitrification activity in response to freeze–thaw cycles among soil types through inspecting soil physical and chemical factors, functional gene expression and enzyme assembly.

## 5. Conclusions

The effects of FTC on denitrification varied across the soil types. Compared with the CK treatment, the soil denitrification potential of FS and LS increased under the FTC and F treatments, while the reverse trend was observed in BS. Soil properties and bacterial communities showed higher similarity between FS and LS than between FS and BS or LS and BS. Microbial composition was altered by FTC treatment, *Gaiella* and Un.Solirubrobacterales were correlated with the denitrification potential in FS and LS. Taken together, our results demonstrated that the influence of FTC on denitrification is strongly dependent on soil type and that the microbial community composition may play a role in this effect.

## Figures and Tables

**Figure 1 microorganisms-11-00593-f001:**
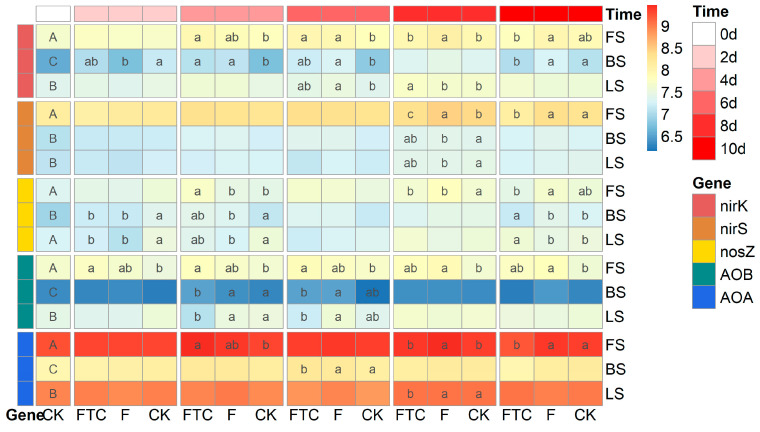
The absolute abundances of the *nirK*, *nirS*, *nosZ*, AOB *amoA* and AOA *amoA* genes (copies g^−1^ dry soil), data were log-transformed; different uppercase letters at 0 d indicate that the gene abundance differed significantly among the three original soils, and different lowercase letters for the same time point indicate a significant difference in the gene abundance between the FTC, F and CK treatments for the same soil type based on Duncan’s test. CK: control treatment (10 °C); F: freezing treatment (−10 °C); FTC: freeze–thaw cycles (−10/10 °C); AOA: AOA *amoA* gene; AOB: AOB *amoA* gene.

**Figure 2 microorganisms-11-00593-f002:**
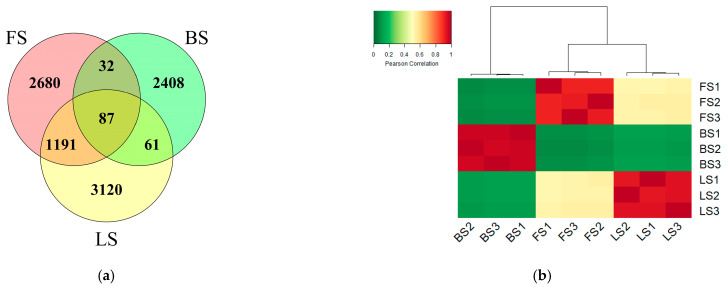
(**a**) Venn diagram showing the distribution of bacterial ASVs in different original soils; (**b**) Heatmap showing the Pearson correlation based on ASV abundances in the three original soils. FS: fluvo-aquic soil; BS: black soil; LS: loess soil.

**Figure 3 microorganisms-11-00593-f003:**
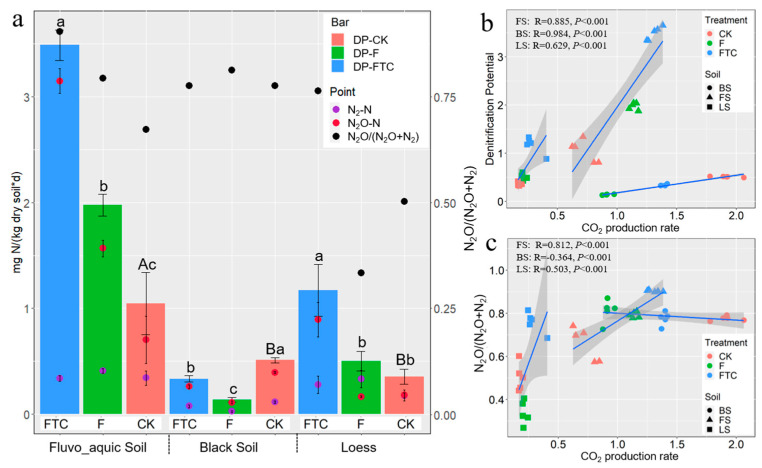
(**a**) Soil denitrification potential (DP) (bar diagram), N_2_-N production rates (purple point diagram), N_2_O-N production rates (red point diagram) and the N_2_O/(N_2_O + N_2_) ratio (black point diagram); (**b**) Relationship between soil CO_2_ production rates and denitrification potential in three treatments from three types of soils; (**c**) Relationship between soil CO_2_ production rates and the N_2_O/(N_2_O + N_2_) ratio in three treatments from three types of soils. Different lowercase letters in the bar chart indicate significant differences in denitrification potential between the FTC, F and CK treatments in the same soil type determined by Duncan’s test; different uppercase letters in the bar chart indicate significant differences in denitrification potential in the CK treatment among the FS, BS and LS. CK: control treatment (10 °C); F: freezing treatment (−10 °C); FTC: freeze–thaw cycles (−10/10 °C); FS: fluvo-aquic soil; BS: black soil; LS: loess soil. Soil DP, N_2_-N and N_2_O-N production rates were the average values of five replicates ± standard error of five replicates.

**Figure 4 microorganisms-11-00593-f004:**
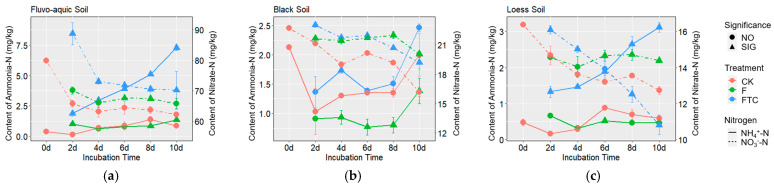
Changes in the soil NH_4_^+^-N and NO_3_^−^-N contents during anaerobic culture of fluvo-aquic soil (**a**), black soil (**b**) and loess soil (**c**). The dotted and solid lines show the variation trends of NH_4_^+^-N and NO_3_^−^-N content, respectively; “SIG” and “NO” indicate significant and non-significant differences in soil NH_4_^+^-N (or NO_3_^−^-N) content between the FTC or F and CK treatment at the same incubation time determined by Duncan’s test; CK: control treatment (10 °C); F: freezing treatment (−10 °C); FTC: freeze–thaw cycles (−10/10 °C).

**Figure 5 microorganisms-11-00593-f005:**
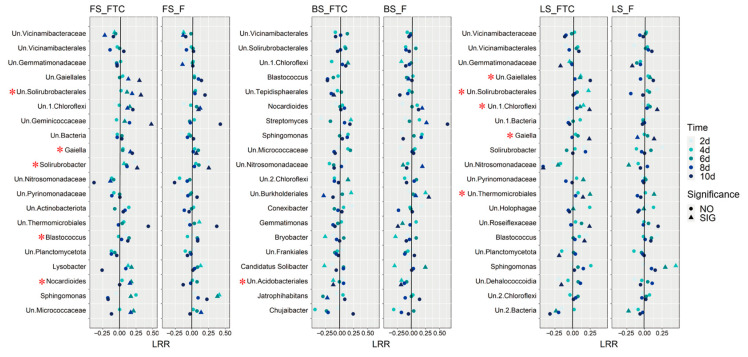
The dot plots show the log response ratio (LRR) of the top 20 genera with the highest relative abundance in the FTC or F treatments compared with the CK treatment at five incubation times in fluvo-aquic soil, black soil and loess soil; “SIG” and “NO” indicate significant and non-significant differences in the relative abundance between the FTC or F and CK treatment at the same incubation time; * indicates a positive Pearson correlation between the denitrification potential and the relative abundance of the genera in the FTC, F and CK treatments at the five incubation times. CK: control treatment (10 °C); F: freezing treatment (−10 °C); FTC: freeze–thaw cycles (−10/10 °C); Un: unidentified taxa.

**Figure 6 microorganisms-11-00593-f006:**
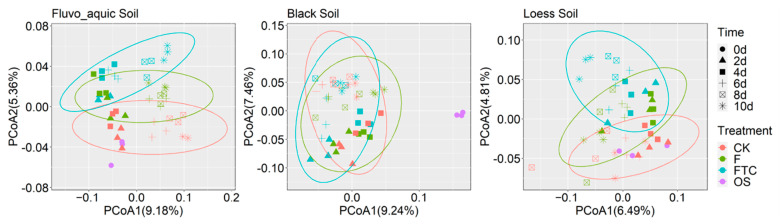
Principal coordinate analysis (PCoA) of fluvo-aquic soil, black soil and loess soil based on the Bray–Curtis distance showing the profile of the bacterial community structure under the FTC, F and CK treatments at different incubation times; CK: control treatment (10 °C); F: freezing treatment (−10 °C); FTC: freeze–thaw cycles (−10/10 °C); OS, original soil.

**Table 1 microorganisms-11-00593-t001:** Soil properties.

Soil	TC mg/g	TN mg/g	C/N	DOC mg/kg	SOC mg/g	DON mg/kg	NO_3_^−^-N mg/kg	NH_4_^+^-N mg/kg	pH
FS	19.52 ^a^	1.69 ^a^	11.54 ^a^	46.44 ^a^	11.44 ^a^	99.97 ^a^	47.84 ^a^	0.35 ^a^	7.94 ^a^
BS	11.75 ^b^	1.28 ^b^	9.20 ^b^	47.23 ^a^	8.47 ^b^	46.92 ^b^	19.56 ^b^	1.01 ^b^	5.40 ^b^
LS	17.38 ^c^	0.89 ^c^	19.32 ^c^	43.98 ^b^	5.34 ^c^	58.79 ^c^	11.40 ^c^	0.47 ^c^	8.25 ^c^

TC, total carbon; TN, total nitrogen; C/N, TC/TN; DOC, dissolved organic carbon; SOC, soil organic carbon; DON, dissolved organic nitrogen; FS, fluvo-aquic soil; BS, black soil; LS, loess soil. Each number represents the average values of three replicates, and the different letters indicate significant differences according to ANOVA with Tukey’s test (*p* < 0.05).

## Data Availability

The raw sequencing data were deposited in the National Center for Biotechnology Information (NCBI) under BioProject ID PRJNA792883.

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
