# Peer review of "Responses of Nitrous Oxide Emissions and Bacterial Communities to Experimental Freeze–Thaw Cycles in Contrasting Soil Types"

_microorganisms, 2023, doi:10.3390/microorganisms11030593_

Round 1
Reviewer 1 Report (New Reviewer)
Overall, this revised manuscript has been substantially improved and could be published after minor revision.
L24: Give the full name of N2O in the first place it appears.
L62: Is that possible that increased C and N substrates due to the broken of physically protected SOM? Then, the interaction of soil C and N turnover. Available C substrate and active microbes would induce soil respiration, and induce soil O2 consumption, which will eventually favour the soil N2O production and emission.
https://doi.org/10.3389/fenvs.2022.1020099
https://doi.org/10.1016/j.scitotenv.2022.158274
Additionally, three soils in this study have various soil physical properties in terms of soil water availability during the thaw, which plays an important role in soil CO2 and N2O emission due to changes in soil gas diffusivity, O2 availability and consumption. Please consider this in the discussion.
https://doi.org/10.1111/ejss.13124
The freeze-thaw cycle has been defined as FTC, better to use it throughout the manuscript.
Figure 2b, c. CO2 subscript.
L472, 480, CO2 and N2O subscript.
Author Response
Please see the attachment.

Reviewer 2 Report (New Reviewer)
I have read the manuscript and in my opinion several corrections need to be made. After that, you can consider publishing this manuscript.
Line 84-86 omit this sentence. The manuscript does not discuss Antarctic soils.
Tabulka 1 Why are statistics deleted? How is the reader supposed to know what is and is not statistically significant? Are they Standard Error or Standard Deviation?
The figure below table 1 does not make sense. What does it belong to? It's unclear. Needs to be corrected.
Fifure 3 It is not clear which of the results are statistically significant. They are just averages. Indicate this fact in the explanatory notes below the figure.
Figure 3 is twice. So what is figure 3???
The discussion is a summary of the results rather than a discussion. This section needs to be corrected.
Author Response
Please see the attachment.

This manuscript is a resubmission of an earlier submission. The following is a list of the peer review reports and author responses from that submission.
Round 1
Reviewer 1 Report
The uthors manuscript describes a thouroughl done experimental study on the effect of freez-Thaw cycles in three different top soils. And oinclude in state of the art level changes of denitrification potentials, gas measurements and bacterial community analyses. ALso analyses of functional genes that reporesent ammonia oxidzers and dentrifiers.
The study revealy differences in DP in reposnse FTC. The major conclusio is that this different reposne can be explained by shifts in taxonmic compostion of the soil bacterial coimunnuity and soil it self.
I am in favor for a publication if the authors adress slef-critically the flllowing lacking information. BActerial Commax has not been assed, but can alos contribute to N"O formation and might explain patterns. Also fungi some soil protist can be denitrifiers thus would contribute to the observed potential DP but were not asses by any functional or bacterial taxonomic gene maker. But it might have been that differetnt fungal abundances a re major reason for the observed different repsonses in the three soils. Moreover, the sudy onyl analysed the top soil (down 10 cm depth) this only a subcompartment of the whole soil and thus has no ecosystem-level implication.
Specific comments to the text
ln 72, better '..to investigate...'
ln 79, better '... , we analysed ...'
ln 86 ff. Since in each country different soil classsification systems exists, It is needed to refer to the FAO classifcation system and deliver some details on soil texture, cropping rotation etc. Thus reader gets a more concrete understanding on the soils self.
ln 99 wrong '..., organic nitrogen (DON), ...' Do you mean dissolved organic nitrogen?
Results
ln 176 ff This hard to read and not needed. I would remove at least the numbers.
ln 177, ln 194-196, ln 204-205, ln 229-230, All these sentences are not needed. They just describe in which figure or table a specific iresult can be found. Such sentences disrupts reading. Please remove them.
Figure 3 c. here you present functional gene makrker absolute abundances. these can only be measured by qPCR. In the M&M section no description of this method including evaulation of potnetial pCR inhibitory effects of DNA extracts has been presented. Please do so.
Conclusions
ln 374 ff is very descriptive. please better resolve (a) what is the limitation of your study (read comments above, (b) what is really NEW, and (c) what can be conclude to improve future studies on the research topic and/or which future question should be answered. PÜlease put a sentence reflecting points a, b c also in the abstract then.
Reviewer 2 Report
The authors have presented data from experiments in which three different soil types have been frozen and thawed, and then emission of nitrous oxide and carbon dioxide have been measured robotically. The authors then isolated total DNA from each sample and assessed changes in the bacterial population from sequential samples. Two conclusions were that denitrification was responsible for most of the nitrous oxide production and some changes in bacterial populations were observed.
This submission was disappointing for two main reasons. First, its value is limited to these specific samples, with little predictive value or mechanistic explanation. Secondly, this reviewer failed to find any statistical analysis of the changes detected, apparently because only single determinations were completed for each soil sample. Consequently, it was unclear whether identical data would be obtained from multiple samples of the same soil type, or even from multiple analyses of the single samples examined. In short, no biological replicates were reported, so it was impossible to judge whether the data are reproducible. The data are therefore anecdotal rather than rigorous. In view of the limited value of the data presented, it would be premature to accept this paper for publication.
Many different factors change rates of nitrification and denitrification is soil: they include, amongst others, water content, temperature, availability of nitrogen, sulphur and phosphorous, presence or limitation of many trace nutrient and especially pH. None of these parameters have been studied.
There is an unfortunate error in the Introduction: NirK and NirS are nitrite reductases, not nitrate reductases!
Reviewer 3 Report
This is a well-written and accomplished article. The study demonstrated that the effects of freeze–thaw cycles on N2O emissions were soil type dependent, and the shift in the community structure of denitrifiers was one factor contributing to the elevated N2O emissions.
An advantage of the article is the use of soils from a long-term field experiment, as well as the use of incubation experiments. The only fundamental comment I have is a request to pay more attention to the relationship between the abundances of N functional genes and process activities. I believe that this is one of the most important tasks in this study. In other aspects, I have a small number of minor comments.
Abstract
L. 27. What is denitrification potential? It it potential denitrification activity ot anything else?
L. 30-32 the abundance of taxa belonging to the Nocardioides genus
Introduction
Please add to the Introduction why it's important to link the resposes of soil bacterial communities and nitrous oxide emissions. Now there is a lack of rationale.
You may use the papers of He et al. 2017, ASE and van Bruggen et al., 2017, SBB which were both discussed the importance of these links.
He, M., Ma, W., Zelenev, V. V., et al. (2017). Short-term dynamics of greenhouse gas emissions and cultivable bacterial populations in response to induced and natural disturbances in organically and conventionally managed soils. Applied Soil Ecology, 119, 294-306.
van Bruggen, A. H., He, M., et al. (2017). Relationships between greenhouse gas emissions and cultivable bacterial populations in conventional, organic and long-term grass plots as affected by environmental variables and disturbances. Soil Biology and Biochemistry, 114, 145-159.
Materials and Methods
L. 86-87. Please provide names of soil types based on the WRB classification as well.
L. 127-128. Were potential or actual denitrification rates measured?
L. 131-132. Please add the information on robotized sampling and analysis systems. For instance, what instrument was used to measure gas concentrations? what detectors?
Results
I would suggest that the authors more clearly and in detail describe the results on the relationships between the numbers of gene copies and the activities of processes in the Results section.
Discussion
L. 355-373. I would argue that the abundances of N-related functional genes were (almost) not related to nitrification and denitrification activities.
Again, I would suggest that the authors more clearly and in detail describe the relationships between the numbers of gene copies and the activities of processes in the Discussion section.